# Robust Identification and Segmentation of the Outer Skin Layers in Volumetric Fingerprint Data

**DOI:** 10.3390/s22218229

**Published:** 2022-10-27

**Authors:** Alexander Kirfel, Tobias Scheer, Norbert Jung, Christoph Busch

**Affiliations:** 1Institute of Safety and Security Research, Bonn-Rhine-Sieg University of Applied Sciences, 53757 Sankt Augustin, Germany; 2Norwegian Biometrics Laboratory, Norwegian University of Science and Technology, 2815 Gjøvik, Norway

**Keywords:** biometrics, fingerprint, optical coherence tomography, OCT, presentation attack detection, PAD, authentication

## Abstract

Despite the long history of fingerprint biometrics and its use to authenticate individuals, there are still some unsolved challenges with fingerprint acquisition and presentation attack detection (PAD). Currently available commercial fingerprint capture devices struggle with non-ideal skin conditions, including soft skin in infants. They are also susceptible to presentation attacks, which limits their applicability in unsupervised scenarios such as border control. Optical coherence tomography (OCT) could be a promising solution to these problems. In this work, we propose a digital signal processing chain for segmenting two complementary fingerprints from the same OCT fingertip scan: One fingerprint is captured as usual from the epidermis (“outer fingerprint”), whereas the other is taken from inside the skin, at the junction between the epidermis and the underlying dermis (“inner fingerprint”). The resulting 3D fingerprints are then converted to a conventional 2D grayscale representation from which minutiae points can be extracted using existing methods. Our approach is device-independent and has been proven to work with two different time domain OCT scanners. Using efficient GPGPU computing, it took less than a second to process an entire gigabyte of OCT data. To validate the results, we captured OCT fingerprints of 130 individual fingers and compared them with conventional 2D fingerprints of the same fingers. We found that both the outer and inner OCT fingerprints were backward compatible with conventional 2D fingerprints, with the inner fingerprint generally being less damaged and, therefore, more reliable.

## 1. Introduction

Over the past few years, biometric methods have been used more frequently in the authentication of individuals. While traditional methods rely on having special knowledge (e.g., passwords) or possessions (e.g., key cards), in biometrics, different physiological characteristics of individuals are observed. Among the various biometric modalities, fingerprint recognition is one of best known and most widely used. Despite the long history behind the usage of fingerprints, there are a number of challenges that limit their use in operational scenarios, e.g., automatic border control systems such as the European Visa Information System (VIS) [1] and the Entry Exit System (EES) [2].

With commercial fingerprint scanners, capturing fingerprints in many non-standard, but nonetheless relevant situations can be problematic. For example, excessively wet or dry skin and skin damage (by injury or disease) can severely impair the quality of fingerprint samples [3,4,5]. In infants, the skin is still very soft, which causes their fingerprint patterns to be squashed when pressed against a sensor surface. This limits the use of fingerprints in applications such as preventing child trafficking or tracking the identity of newborns in hospitals [6,7]. Furthermore, many fingerprint systems are not resistant to presentation attacks, in which an impostor presents artifacts of fingerprints with the intent to impersonate the target victim [8]. The recognition of such attacks is called presentation attack detection (PAD) and remains an unsolved challenge for many attack scenarios. Secure PAD-enabled fingerprint systems usually require several additional single-purpose sensors for decision-making [9].

A promising solution to the mentioned problems is be the use of optical coherence tomography (OCT). With an OCT scanner, the fingerprint capture process can be extended into the third dimension, which means this approach could even provide an intrinsic PAD solution without the need for additional sensors. Furthermore, the identification of individuals in difficult situations could potentially be improved by using subsurface skin features, making it more difficult or even impossible to successfully perform a presentation attack with today’s artifact types. The aim of this work was, as a first step, to develop a method for efficient segmentation of fingerprints from volumetric OCT scans using standard PC hardware.

## 2. OCT Fingerprinting

### 2.1. OCT Basics

Optical coherence tomography (OCT) is a noninvasive imaging technique, which is often referred to as the optical equivalent of ultrasonic pulse echo imaging. The method uses low-coherence interferometry to capture depth-resolved images from within optical scattering media (e.g., biological tissue) based on optical delay. While the effective imaging depth is typically limited to less than one millimetre, the spatial resolution can be as high as a few micrometers [10]. Most OCT implementations provide cross-sectional images, called B-scans, by acquiring a series of axial measurements, called A-scans, and combining them into one image. Similarly, a full-volume scan is created by capturing and stacking together individual B-scans. The term en-face images or C-scan refers to an image created by slicing the volume horizontally at a certain depth.

While there are many different implementations, all are based on the principles of the Michelson interferometer: A low-coherence beam of light is divided by a beam splitter into two beams, one directed to a mirror (the reference arm) and the other to the measured object. After reflection, the beams recombine at the (same) beam splitter, where they cause interference, which is picked up by a sensitive photodetector. Since interference can only occur if the light has travelled a similar distance in both interferometer arms, this setup can be adjusted very precisely to a desired measurement depth.

Generally, there are two fundamental OCT techniques: time domain (TD) and Fourier domain (FD). Figure 1a shows the traditional TD variant, which was first demonstrated by Huang et al. in 1991 [11]. In this method, the axial resolution corresponds to the coherence length of the light source and the scan depth is controllable via the positions of the reference mirror. Since only one point can be captured at a time, each A-scan requires multiple consecutive measurements. Capturing a full-volumetric image, therefore, requires mechanical scanning in all three directions, which would be too slow for real-time fingerprint recognition.

FD-OCT, on the contrary, measures the wavelength-dependent reflectivity using a broadband light source to observe a multitude of different wavelengths, as each wavelength provides information on the strength of the periodic refractive index modulation (Fourier component). By applying a Fourier transformation to the reflectivity, expressed as a function of the wave-vector, a complete A-scan is obtained from a single measurement. In this case, the axial range corresponds to the coherence length of the light source [10]. FD-OCT has two competing designs: spectral domain-OCT (SD-OCT) and swept source-OCT (SS-OCT). While SD-OCT, depicted in Figure 1b, has a broadband light source and a spectrometer to measure the spectrum, SS-OCT combines a single detector with a tunable narrow-band light source, which can sweep rapidly over the required optical bandwidth. Both approaches have in common that the reference mirror is static, which leads to the obvious speed advantages, as there are less moving parts. However, the key feature of FD-OCT over TD-OCT is the improved sensitivity as a result of the material reflectivity detection as a function of wavelength [10].

While the previously described OCT modalities measure single depth profiles, there are other potential OCT variants that could directly produce cross-sectional images. Line field-OCT (LF-OCT) is an FD variant that can capture entire B-scans in a single measurement [12]. For this purpose, the light beam from a tunable narrow-band light source is formed into a line, and a line-scan camera is used as a detector. To capture an entire volume, the scanning beam only needs to be moved in one direction over the sample. Full-field-OCT (FF-OCT) is a TD variant that captures the en-face images by using a camera (2D matrix sensor) instead of a single photo detector [13]. For a complete volume scan, only the reference mirror has to be moved. Unfortunately, this variant suffers from long acquisition times due to the mechanical movement of the reference arm and the oversampling required to compensate for the low signal-to-noise ratio as a result of measuring in the time domain.

### 2.2. Anatomy of Fingertip Skin

The protruding lines on the skin surface of fingertips, called friction ridges, have long been used to authenticate individuals. Features extracted from the biometric characteristic fingertip are hierarchically organized into three levels: Level 1 refers to the general ridge flow pattern and type; Level 2 (Galton points) includes discontinuities such as ridge endings and bifurcations; Level 3 (shape) adds dimensional attributes such as ridge width, shape, and sweat pores. As these features become smaller at higher levels, more sophisticated sensors must be used to capture them. For the first two feature levels, the de facto standard for fingerprint capture devices demands a resolution of 500 ppi, whereas for Level 3, at least 1000 ppi is recommended [14].

The outer part of human skin can be roughly subdivided into two layers: epidermis and dermis. The surface of the epidermis represents the commonly used epidermal, surface, or outer fingerprint. The interface between the two layers, called the epidermal–dermal junction, consists of connective tissue arranged in double rows. In biometrics, this region is called the dermal, subsurface, or inner fingerprint because it serves as a template for the outer fingerprint. Newly formed skin cells migrate from this region to the skin surface, where they constantly renew the outer fingerprint. This allows the outer fingerprint to recover from superficial injuries that do not extend into the inner fingerprint region. The analysis of the inner fingerprint was discussed by Plotnick as early as 1958 [15], but at that time, there was no way to capture the inner fingerprint in vivo. With the advent of OCT, it became possible to observe the inner fingerprint in live fingers at sufficient resolution. Due to differences in the optical density of the individual skin layers and the air, the fingerprints appear brighter than the rest of the image, see Figure 2a. The full volume scan can be visualized using ray-marching, see Figure 2b.

## 3. Related Work

Observation and analysis of the specific anatomy of the fingertip skin beyond the outer skin layer is rarely addressed in the biometric literature. This can be explained by the currently still high cost of suitable OCT acquisition equipment, which is why most of the work in this field has been carried out in funded projects at research institutes.

The first known work addressing the use of OCT in fingerprint biometrics was published in 2006 by Cheng and Larin [16,17]. They showed that thin-film forgeries could deceive commercial fingerprint scanners, but were detectable with an OCT scanner. In this early study, a TD-OCT scanner was used, which could capture only a small fraction of the fingerprint and still took several seconds to acquire the data.

Darlow et al. published several papers at the South African Council for Scientific and Industrial Research (CSIR) in 2015 and 2016. First, they evaluated various digital filters for speckle reduction [18], which is a multiplicative noise component, which cannot be avoided in the OCT scanning process. Later, they published an algorithm that can find and extract the inner fingerprint in OCT scan data [19,20]. Furthermore, they created a hybrid fingerprint by fusing the internal and external components [21,22]. This allowed them to reconstruct damaged areas in the outer fingerprint with information from the inner one. Automatic PAD [23], as well as the identification of individuals with skin diseases using the inner fingerprint have also been discussed [24].

The EU project INGRESS, which ended in 2017, focused on capturing the inner fingerprint with a custom-built full-field (FF)-OCT scanner. Since this OCT variant directly produces C-scans (en-face images), the finger had to be pressed against a glass pane to flatten and stabilize it. The applied method generated an image of the internal fingerprint based on multiple measurements over a small depth range, which were then averaged to produce the final result. Auksorius and Boccara of the Institute Langevin developed two different FF-OCT scanners during this project. The first device paired an InGaAs camera and NIR illumination with a wavelength of 1300 nm [13], which is well suited for skin penetration. However, this specific camera technology is notoriously slow, has a low resolution, and is very expensive. The associated costs were the main reason for the second development, which used a (standard) silicon camera and 900 nm illumination [25,26]. By modifying the system setup, a higher frame rate and resolution were achieved while reducing the cost to only a fraction of the first system. The final setup was able to produce 17 mm × 17 mm en-face images of the inner fingerprint in just 0.3 s. This impressive result was possible because only a few en-face images were needed. However, the depth of the internal fingerprint is known to vary between individuals, which is an issue that was not discussed. It should also be noted that, for full-volume scans, FF-OCT has significantly longer acquisition times than the conventional scanning-type OCT variants used by most other research groups.

The German Federal Office of Information Security (BSI) conducted several projects on OCT fingerprinting, and recent work was supported by the Norwegian Biometrics Laboratory (NBL) at the Norwegian University of Science and Technology (NTNU).

An early study confirmed the general suitability of OCT for imaging the inner finger structure, as well as the sweat ducts between the outer and inner fingerprint [27]. Furthermore, it was shown that, not only the presence of thin-film artifacts, but also the underlying skin structure of the attacker can be detected. Based on these preliminary investigations, Sousedik et al. proposed an efficient algorithm for boundary detection between layers in an OCT fingerprint scan [28]. Since the algorithm tends to generate outliers, they further developed their method by using a neural network to represent the fingerprint surface and deal with these outliers [29]. They also derived quality metrics based on the integrity of the boundaries, since the presence of outliers is often directly related to the non-compliant behaviour of the captured person (e.g., finger movement during scanning).

In a follow-up project, an SD-OCT scanner was developed specifically for fingerprint biometrics [30,31]. With its large scan area of 20 mm × 20 mm, the device can capture an entire fingerprint in up to 1408 × 1408 × 1024 voxels at a fixed 100 kHz line rate. For this, the scan head was mounted facing upwards, and a palm rest was installed to keep motion-related distortions in the OCT data at an acceptable level. Another challenge faced was the time-efficient processing of the gigabyte-sized datasets, from which all relevant information had to be extracted within just a few seconds. This problem was later addressed at NTNU, where Sousedik et al. developed a novel edge detection algorithm that can be very efficiently executed on GPUs [32]. We therefore considered the BSI and NBL research on OCT fingerprinting as the starting point for our own work.

## 4. OCT Capture Device

Prior to the actual research tasks, a suitable measurement setup had to be designed. Since our main focus was on the development of suitable algorithms, a commercial OCT scanner was procured from Thorlabs, a well-known optics manufacturer. The main components include a base unit (TEL320), galvo-mirror scan head (OCTG-1300), and telecentric scan lens (OCT-LK4). To enable a familiar scanning process as with conventional fingerprint capture devices, we decided the OCT scanner should see the fingertip from below. The fingertip should, however, not be pressed against an optical window (e.g., a glass plate) to avoid deformation of the skin. Therefore, we developed a finger rest that suspends the fingertip above the scan head, which is mounted in an upside-down orientation on a linear actuator with micrometre accuracy. In this way, the measurement distance can be precisely controlled by the software to align the fingertip with the top edge of the scan volume for optimum image quality. The sensitive and heavy components were integrated into a custom-built rolling container (see Figure 3) for protection, while maintaining a degree of mobility. The OCT setup has the following characteristics:Centre wavelength: 1300 nm;Field of view: 16 mm × 16 mm;Imaging depth: 3.5 mm (air)/2.6 mm (water);Resolution: 20 µm (lateral) × 5.5 µm (axial);Scan size: 1024 × 1024 × 1024 voxels;Scan rate: 146 kHz line rate;Scan time: ca. 8 s (for this scan size).

## 5. Fingerprint Segmentation

This section describes the main topic of this work, which is the segmentation of the outer and inner fingerprint.

### 5.1. The Outer Fingerprint

The outer fingerprint is located at the highly reflective air–skin boundary, which appears as a bright thin layer in the volume scan. Since this layer is typically the most clearly visible feature of the scan, it is relatively easy to find despite its curved shape. Our fingerprint segmentation algorithm (see Section 5.3) can detect the outer fingerprint directly in the full-volume scan, which it then outputs in the form of a heightmap, hereinafter referred to as surface. Since the position of the finger is initially unknown, this surface covers the entire scan area, not just the fingertip. We therefore needed to post-process the surface using the fingerprint masking algorithm (see Section 5.4), which removes the invalid data points.

### 5.2. The Inner Fingerprint

Due to signal degradation and increased noise below the skin surface, the inner fingerprint is usually much more difficult to recognize than the outer fingerprint. Our solution to this problem was to move the scan lines independently towards the top edge of the volume, according to the position of the outer fingerprint, while discarding all data points beyond the volume boundaries. This flattening brought the dermal–epidermal junction closer to the top of the volume and allowed for the truncation of the outer fingerprint from the dataset, which would otherwise interfere with the inner fingerprint when we run the fingerprint segmentation algorithm a second time. However, we found that an algorithmic flattening of the dataset can transfer features of the skin surface to the lower skin layers, resulting in a kind of hybrid fingerprint, which could impact the reliability of PAD. For this reason, we propose to flatten the volume according to Equation (Equation 1) against the outer fingerprint envelope $E_{S}$ provided by the fingerprint envelope algorithm (see Section 5.5). In this, ΔS denotes the height (thickness) of the outer fingerprint layer, ensuring its complete removal. Finally, the inner fingerprint surface is unflattened by reversing the shifts, restoring its real shape and position in the volume.
(1)$$\begin{array}{cc} \bar{V}\left(x,y,z\right)\; & \leftarrow \;V(x,y,z\;+\;E_{S}\left(x,y\right)\;+\;\Delta S)\;,\;\forall x,y,z\\ \Delta S\; & :=\;max\{S\left(x,y\right)-E_{S}\left(x,y\right)\} \end{array}$$

Figure 4 gives an overview of how the algorithms are chained together. Their operation principle is explained in the following subsections.

### 5.3. Algorithm: Fingerprint Segmentation

#### 5.3.1. Fast and Robust Edge Detection

Inspired by the work of Sousedik et al. [32], we tried to identify the fingerprints at the boundaries between the different layers in the scan observed along the scan lines. For this, we applied a custom 1D edge detection filter with additional low-pass characteristics for noise suppression individually to each scan line. The size of the filter’s convolution kernel is adjustable, which enables support for different scan resolutions and allows the filter to adapt to variations within a scan. The optimal kernel size depends on the height of the skin layers and can therefore reach a significant fraction of the scan line length. Since it would be very inefficient to perform convolution directly with this kernel, it was derived twice, resulting in a sparse representation (Equation (Equation 2), Line 3), which, regardless of its size, has exactly four non-zero filter taps and is therefore much faster to process.
(2)$$\begin{array}{c} \begin{array}{ccccccccccccccc} \kappa_{L}^{\left(0\right)}=[ & 1 & 2 & 3 & 4 & \cdots & \left(L/2\right) & \; & \left(-L/2\right) & \cdots & -4 & -3 & -2 & -1 & ]\\ \kappa_{L}^{\left(1\right)}=[ & 1 & 1 & 1 & \cdots & \; & 1 & \left(-L\right) & 1 & \; & \cdots & 1 & 1 & 1 & ]\\ \kappa_{L}^{\left(2\right)}=[ & 1 & 0 & \cdots & \; & 0 & \left(-L-1\right) & \; & \left(L+1\right) & 0 & \; & \cdots & 0 & 1 & ] \end{array} \end{array}$$

To facilitate the comparison of different kernels based on their absolute filter response, their normalized form was used instead, which ensured that, when applied to the step function, the peak response would be the same for any kernel size:(3)$$k_{L}^{\left(i\right)}=\kappa_{L}^{\left(i\right)}\times \frac{2/L}{L/2+1}\;,i=0,1,2$$

With this, the convolution (*) of the original kernel and a single scan line *s* can be performed very efficiently based on Equation (4) in which the double-integral cancels the second-order derivative, while the zero- and first-order derivatives contribute only the integration constants. Since most operations are performed on the sparse kernel, the total computational cost is almost independent of the kernel size.
(4)$$\begin{array}{cc} \left(k_{L}^{\left(0\right)}*s\right)\left(z\right) & =\int_{z_{0}}^{z}db\int_{z_{0}}^{b}\left(k_{L}^{\left(2\right)}*s\right)\left(a\right)da\\ \; & +\left(k_{L}^{\left(1\right)}*s\right)\left(z_{0}\right)\times \left(z-z_{0}\right)\\ \; & +\left(k_{L}^{\left(0\right)}*s\right)\left(z_{0}\right) \end{array}$$

The filter was optimized for particularly fast execution on the GPU by processing each scan line in a separate GPU thread. To avoid costly caching of intermediate results in off-chip memory, integration and convolution are performed simultaneously, requiring only a single pass over the scan line. As part of the surface detection (Section 5.3.6), the filter was processed by our test system in about 40 ms per one million voxels stored in a 32-bit floating point, which is the native output format of our OCT scanner. For comparison, the same algorithm executed more than an order of magnitude slower on the CPU. To further improve GPU performance, we decided to cast the data to unsigned 8-bit integers, since their value range covers the full dynamic range of the scan data, losing only the decimal places. We found this conversion to be worthwhile, as it improved memory consumption by 75%, data transfer time by 50%, and overall execution time by 25%, without significantly affecting the quality of the resulting fingerprint. There was no additional delay, as the conversion can be performed in parallel with the capture process using vacant CPU resources.

#### 5.3.2. Intensity Roll-Off Compensation

An inherent limitation of the OCT system used is the depth dependence of the sensitivity. This means that, when capturing curved objects, such as a fingertip, the intensity profile is different for regions that are closer to the scanning head than for those farther away. As a consequence, the outer parts of the fingerprint have lower intensity and contrast, which makes segmentation of the fingerprint surface more difficult here. To mitigate this problem, the volume intensity should be normalized. For this, we calculated the average per-depth intensity for the entire volume and ran a linear regression to measure the intensity decline, which was then used to artificially amplify the voxel values at the deeper layers.

#### 5.3.3. Volume Pyramid

Segmenting the fingerprint surface based on the position of the maximum filter response in each scan line would result in a very rough surface full of holes and peaks. This is because edge detection is inherently susceptible to noise, of which there is plenty in an OCT scan. To address this problem, we propose the use of a multi-resolution volume pyramid for noise reduction, created by downsampling the scan volume multiple times in all three dimensions. Sousedik et al. used a similar approach, but performed the downsampling only in the horizontal plane and left the depth unchanged [32]. After implementing and comparing both approaches, we concluded that downsampling in all dimensions as the better alternative because it does not deform the finger by changing the aspect ratio. Moreover, it was also about 15% more time and memory efficient.

#### 5.3.4. Contrast Enhancement

Occasionally, the edge detection filter may have problems locating the fingerprint surface in poor quality scans, e.g., due to suboptimal placement of the finger on the scanner. To mitigate this problem, the volume pyramid was preprocessed to improve contrast at the higher resolutions. Starting with the most downsampled volume $V_{N}$, the volumes $V_{i}$ were successively upsampled and then voxelwise multiplied with their respective predecessor volume $V_{i-1}$, and the result was normalized to avoid integer overflow. The basic idea of contrast enhancement is that, up to a certain level of downsampling, the macrostructure of fingerprints can be represented more reliably by downsampled voxels than by their higher-resolution counterparts. Therefore, embedding this information in the higher resolution volumes increases the likelihood that the fingerprint surface will be correctly identified in low-contrast situations without sacrificing accuracy when the enhancement would not be required. It should be noted that, when upsampling $V_{i}$ to $V_{i+1}$, a simple doubling or averaging of values can trigger the formation of clusters, creating artificial edges in the high-resolution volumes. To resolve this problem, we used trilinear interpolation to resample $V_{i}$ at positions that each correspond to voxel coordinates in $V_{i+1}$ and are therefore slightly offset from the voxel centre in $V_{i}$.

#### 5.3.5. Fingerprint Region

Due to the high amount of noise in OCT data, edge detection results can only be trusted within a relatively narrow band, called the (fingerprint) region *R*, which is centered around the expected true fingerprint surface. Since the position of this surface is initially unknown, the region is set to encompass the entire volume. From there, it is incrementally narrowed down as Algorithm 1 surface segmentation traverses the volume pyramid towards full resolution.
**Algorithm 1:** Fingerprint segmentation   **Data**: Volume *V*, Mask *M* (optional)   **Result**: Surface *S*   Def. $f_{w}\;$ := region scaling factor (hardware dependent)   Def. $w_{N}$ := initial search region size (hardware dependent)   Def. $L_{N}$ := set of initial kernel sizes (scan size and resolution dependent) **1**   Copy the full volume scan *V* from CPU memory to $V_{0}$ located in GPU memory   ▹Intensity roll-off compensation: **2**   Store the average intensities of the en-face slices $V_{0}\left(:,:,z\right)$ as a vector A(z) **3**   Run a linear regression on A(z) to measure the deviation δ(z) from the average  **4**   Set $V_{0}\left(x,y,z\right)\leftarrow V_{0}\left(x,y,z\right)-\delta \left(z\right)\;\;\forall x,y,z$ to normalize the intensity profile   ▹Volume pyramid and contrast enhancement: **5**   Generate N>0 additional versions of $V_{0}$ where $V_{n+1}$ is the result of downsampling $V_{n}$ by a factor of two along all three dimensions (note: for upsampling and downsampling, the GPU’s texture mapping unit is used in linear interpolation mode)

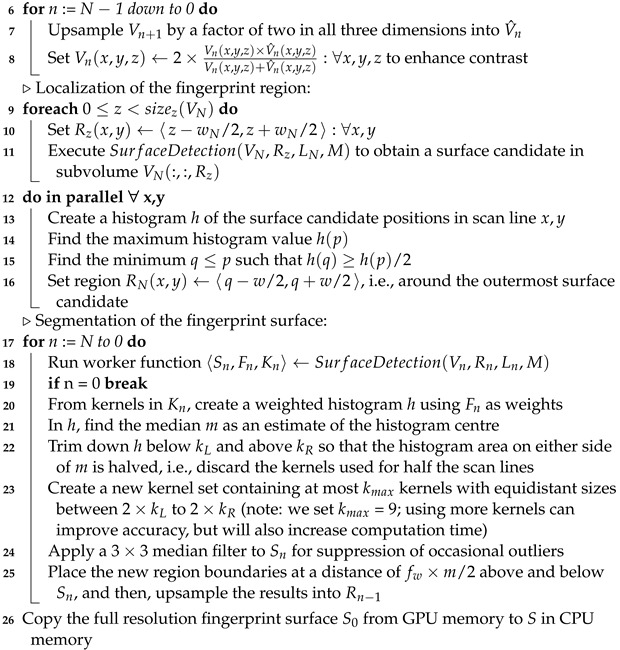


In non-contact fingerprint scanning, region size varies greatly between individual fingers and can therefore not be predetermined. The wider the region, the higher the risk of noise-related outliers in the surface. The narrower the region, the less margin there is to correct such outliers at a later stage. In both cases, the segmentation result are likely suboptimal. For this reason, we optimize the region by setting its size to a resolution-specific percentage of the weighted median kernel size used in a previous surface detection run (Algorithm 2), where the weight is defined as the maximum filter response within the previous region. The rationale for this approach is that kernel size usually correlates with skin layer thickness, which means the derived region is neither too narrow nor does it extend into the adjacent skin layers.
**Algorithm 2: **Surface detection   **Data**: Volume *V*, region *R*, kernels *L*, mask *M*   **Result**: Result Set RS

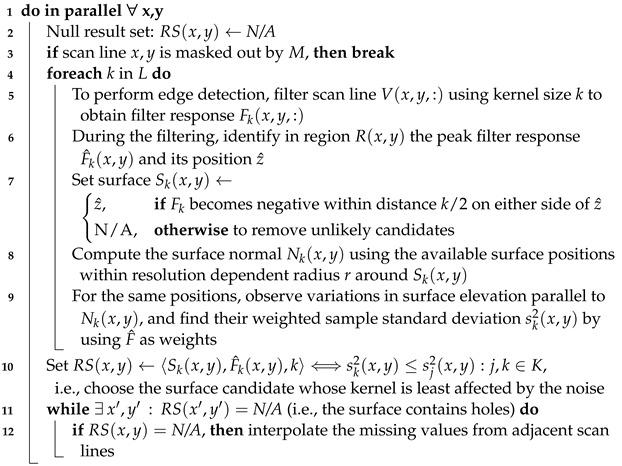


#### 5.3.6. Surface Detection

As the thickness of the skin layers varies from person to person and also within the same finger, using a single kernel size for the entire fingerprint does not guarantee good results. Therefore, we need to identify the optimal kernel size for each scan line separately. For this purpose, Algorithm 2 filters the volume using a set of equidistantly sized kernels and identifies the maximum filter response for each scan line within the fingerprint region. As the algorithm inevitably encounters many unsuited kernel sizes, the position of the maximum filter response is considered a potential surface position only if the response declines to zero within the kernel boundaries on either side of the maximum filter response. This condition increases the likelihood that false edges are discarded for mismatched kernels. To select an optimal surface position, we have to make two assumptions: Firstly, we assume that skin layer thickness does not change rapidly throughout the fingertip, which means there should be at least one matching kernel for each scan line that also performs well in adjacent scan lines. Secondly, the scan resolution is sufficiently high for the surface curvature to be negligible in small areas. To assess the reliability of each kernel at any given point on the fingertip, we calculate the weighted standard deviation from the surface positions provided by the same kernel within a small radius, using the peak filter response as weight. The optimum surface position is then selected based on the kernel that produces the lowest variance. We found that probing the scan data using various kernel sizes results in much more reliable approximation of the fingerprint surface, especially in low-quality scans. We found that probing the scan data locally with different sized kernels leads to a much more accurate approximation of the true fingerprint surface than using the same kernel globally, most notably in low quality scans. While the surface may still contain holes in the places where none of the kernels has worked, these holes are easily collapsed by repeated interpolation at their perimeter.

### 5.4. Algorithm: Surface Masking

A segmented fingerprint surface may spuriously extend beyond the fingertip, provided the scan area was not completely covered by the fingertip. Initially, we attempted to isolate the fingertip based on general assumptions about the shape and orientation of the finger. However, this approach failed with unfavourably shaped fingers or whenever noise led to bad segmentation results that resembled a fingertip. We therefore propose a shape-independent masking approach that relies solely on the voxel intensities.

Algorithm 3 compares the respective maximum voxel values above and below the fingerprint surface taken from the outer fingerprint region. Since the external fingerprint is detected exactly at the air–skin boundary, the values from below the surface are typically much larger than the values from above the surface for only the scan lines covered by the fingerprint. This means that, while most positions inside the fingerprint were classified correctly, initially, about half of the outside positions were classified incorrectly. To remove these errors, we blurred the mask using a box filter whose kernel size was set to the expected pixel distance between the friction ridges. The resulting grayscale image was then re-binarized using an adaptive thresholding technique (Otsu’s method). At this point, there may be only a few isolated spots left that have not yet been assigned the correct mask value. We detected and eliminated them by using a series of flood fill operations to determine the contiguous outer contour of the fingerprint, whose edges are smoothed for the final result.
**Algorithm 3: **Surface masking  **Data**: Volume *V*, outer fingerprint surface *S*  **Result**: Mask *M*  Def. $w_{n}$ := outer fingerprint region size

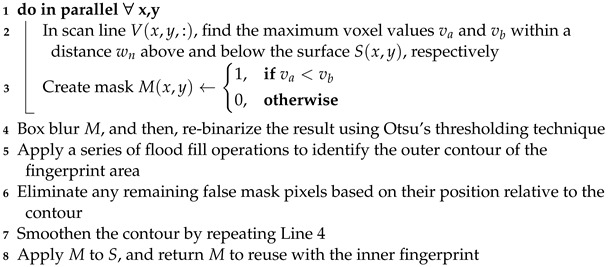


### 5.5. Algorithm: Surface Envelope

Flattening the volume against the outer fingerprint would lead to undesirable distortions in the skin layers, which could corrupt the inner fingerprint. We therefore need a variant of the fingerprint surface that retains the general shape of the finger but does not contain information about the friction ridges. For this purpose, Algorithm 4 computes two additional surfaces called the upper and lower surface envelope. While the upper touches the fingerprint at its ridge tops, the lower only grazes its valley bottoms. The envelope is computed separately for each B-scan using an exhaustive search of the ridge tops, starting at the highest ridge (i.e., the highest point on the fingerprint surface) from where other ridge tops are found one-by-one in either direction. Observed from the current top, the next top is observable on the horizon of the (generally) downwards curved surface. However, since not all fingertips have a perfectly convex shape, the visibility range is limited to the expected distance between ridges. A preliminary envelope surface is then created by filling in the missing positions between adjacent tops using linear interpolation. This surface is not yet reliable, as it can dip into the valleys wherever the friction ridges are crossed at too shallow an angle. For this reason, the same process is repeated in the orthogonal direction, creating a second surface whose errors are expected to not coincide with the first. The envelope is then finalized by selecting the points from both preliminary surfaces that are closest to the topside of the scan. Lastly, a simple box filter is applied to remove the surface roughness caused by diagonal ridges or skin imperfections. The same algorithm is also used to compute the lower envelope, as needed later for the creation of 2D fingerprints. This is done using a version of the fingerprint surface that has been vertically mirrored at the upper envelope to invert its peaks and valleys, and then the resulting lower envelope is flipped back to its normal orientation.
**Algorithm 4: **Surface envelope   **Data**: Surface *S*, mask *M*   **Result**: Envelope $E_{up}$, $E_{low}$   Def. *l* := anatomy- and resolution-dependent constant set to the expected maximum distance between adjacent papillary ridges

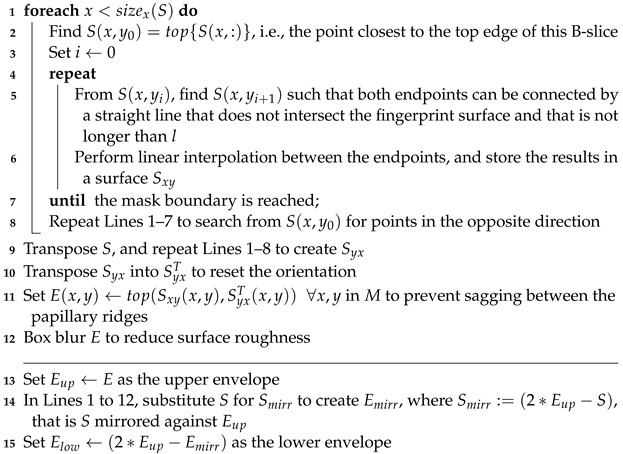


### 5.6. The 3D-to-2D Conversion

The 2D fingerprints presented here are the result of a simple parallel projection of their 3D surface, Equation (Equation 5). For this, the surface is flattened against its upper envelope. The remaining surface undulations are then normalized based on the distance between the upper and lower envelope to compensate for natural variations in ridge height. Any surface positions outside the unit interval, resulting from imperfections in the envelopes, are simply clamped to the interval endpoints. The resulting image shows the ridges in darker shades than the valleys.
(5)$$\begin{array}{cc} FP2D\; & \leftarrow \;clamp\left(\frac{S-E_{up}}{E_{low}-E_{up}},\;min=0,\;max=1\right) \end{array}$$

## 6. Evaluation

### 6.1. Segmentation Results

Figure 5a shows a single B-scan picked from the middle of a typical fingerprint scan. In Figure 5b the same image is overlaid with cross-sections of the segmented fingerprint surfaces and their respective envelopes. Figure 6 shows the entire 3D surfaces (Figure 6a,b) and their respective 2D representation (Figure 6c,d). As can be readily seen, the outer and inner fingerprint share the same ridge pattern. In this particular case, the outer surface is in a less pristine condition than the inner surface. The inner surface appears to be unaffected by superficial skin damage (scratches) and adhering dirt particles. However, the inner fingerprint is not as clearly delineated as the outer fingerprint, most notably towards the edges, where intensity roll-off (Section 2.1) becomes increasingly pronounced.

Upon close inspection, the double-rowed ridge tops (Section 2.2) can be seen in the inner fingerprint, which are rounded in the outer fingerprint. Both fingerprints can be binarized and their minutiae points extracted; see Figure 6e,f, which results in a near-perfect match for most healthy fingers.

### 6.2. Runtime Evaluation

For OCT fingerprinting to become a useful complement to conventional methods, the response time should be as short as possible. Operational requirements define a maximum of 10 s for practical applications [33,34]. This time is mostly spent on the long acquisition process, which lasts about 8 s, leaving 2 s for data processing. The desired quality of the results was achieved on normal consumer hardware, with most of the processing performed by GPU computing. The execution times are summarized in Table 1. Since we were able to undercut the target, there is room for future extensions to the processing chain.

### 6.3. Compatibility with 2D Fingerprints

To assess the validity of our method, we performed a *N*:*N* comparison between OCT fingerprints and plain 2D fingerprints. For this, we captured 130 fingers (13 subjects, 10 fingers each) using our OCT device and a commercial Dermalog LF10 fingerprint scanner. A commercial fingerprint identification software Verifinger 12.3 was used for the comparison. The metrics considered were mated comparison trials (FNMRs) and non-mated comparison trials (FMRs). The equal error rate (EER) expresses the percentage of misclassifications at the cross-over point of the FMR and FNMR curves. The failure-to-extract (FTX) metric indicates the percentage of fingerprints for which the comparison software was unable to extract features.

Comparing the outer fingerprints to the 2D fingerprints resulted in an EER of 5.5% and an FTX rate of 1.5% (Figure 7a). For the inner fingerprints, the numbers improved to an EER of 2.7% and an FTX rate of 0.7% (Figure 7b). For comparison, the 2D fingerprints had an FTX rate of 0.6%.

Manual inspection of the fingerprints revealed severe skin abrasions and scratches on many of the outer fingerprints. The inner fingerprints, on the other hand, were completely intact. Therefore, it is not surprising that they performed better than their outer counterparts.

### 6.4. Compatibility with Foreign Data

To further test the robustness of our method, we processed OCT fingerprints from the OCT II project. Their data size is twice as large as ours, at 2 GB per scan. Since the computation time increases linearly with data size, it doubled to about 1659 ms. Despite the differences in the data size, field of view, and scan wavelength of their custom-built OCT device (see Section 3), our method still worked as intended without any modifications. We therefore have a high degree of confidence in the robustness of our method.

## 7. Conclusions and Future Work

In this work, we presented a complete digital signal processing chain for segmenting fingerprints from OCT scans. Our GPU-optimized solution is capable of processing a gigabyte-sized fingertip scan in less than one second using standard PC hardware. The segmented OCT fingerprints have sufficient quality for minutiae point extraction and are backward compatible with conventional 2D fingerprints. Our studies showed a high resilience of the inner fingerprint against superficial skin damage. While the outer fingerprint can be severely compromised, the inner fingerprint is usually not affected, which gives OCT a decisive advantage over conventional acquisition methods. In our future studies, we intend to investigate this in more depth. To this end, we plan to improve the 3D-to-2D mapping to emulate the traditional touch-based capture for which the existing fingerprint matching devices are designed. Furthermore, we aim to develop an inherently secure PAD mechanism that leverages the presence of the inner fingerprint and additional skin features (e.g., sweat ducts) missing from today’s artifact fingers.

## Figures and Tables

**Figure 1 sensors-22-08229-f001:**
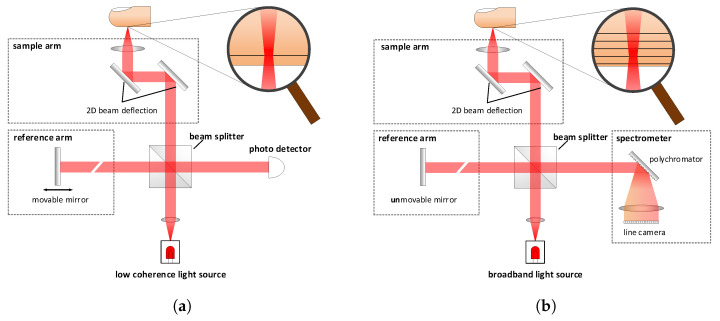
Basic OCT setup. (**a**) Time domain. (**b**) Spectral domain.

**Figure 2 sensors-22-08229-f002:**
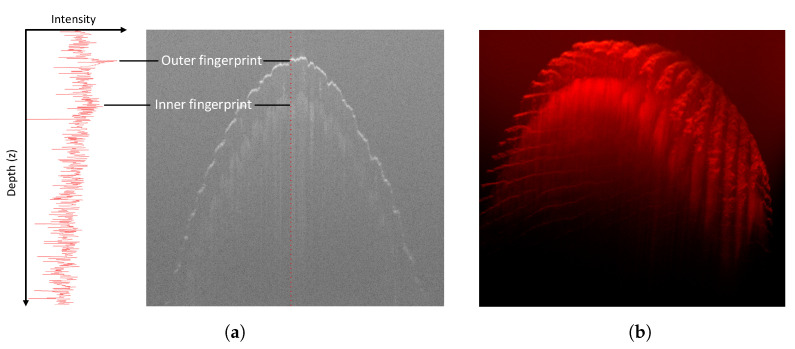
OCT scan data. (**a**) A-scan (left) and B-scan (right). (**b**) 3D scan rendered with ray-marching.

**Figure 3 sensors-22-08229-f003:**
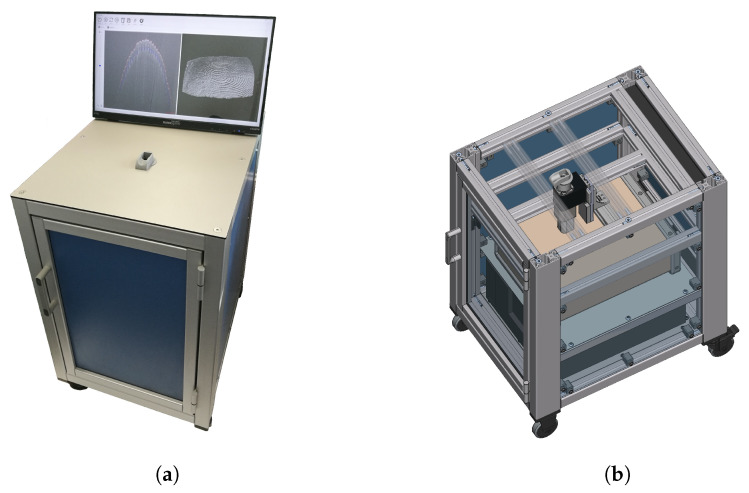
OCT container setup. (**a**) Running our measurement and processing software. (**b**) CAD model showing the scan head mounted to a linear stage.

**Figure 4 sensors-22-08229-f004:**
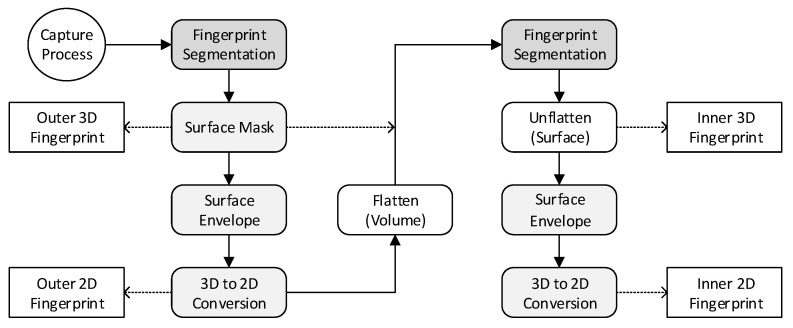
Complete signal processing chain.

**Figure 5 sensors-22-08229-f005:**
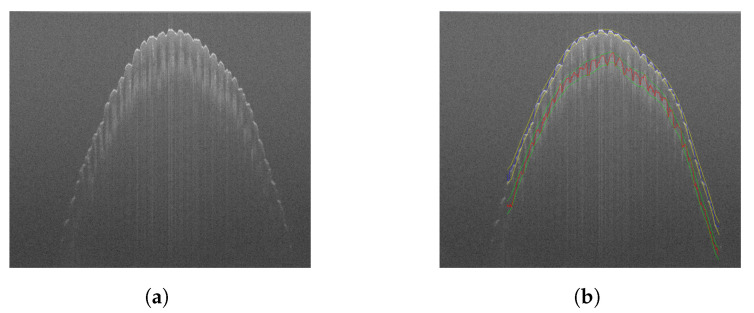
Segmentation results viewed on a B-scan. (**a**) Raw B-scan for reference (**b**) Overlaid with outer and inner fingerprint surfaces and envelopes.

**Figure 6 sensors-22-08229-f006:**
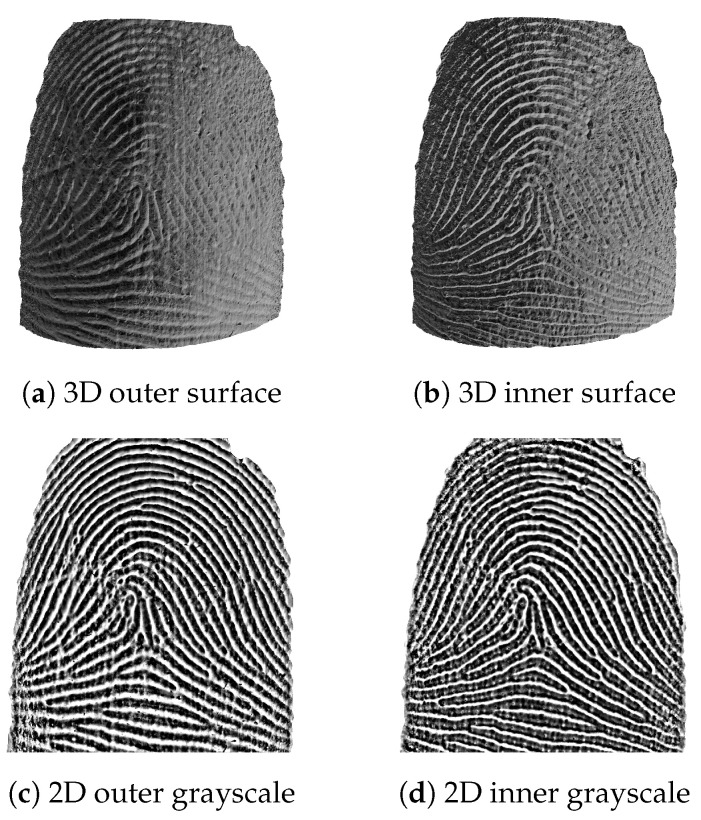
Segmentation results for the same finger: outer (**a**) and inner (**b**) fingerprint surface, their respective 3D-to-2D conversion result (**c**,**d**), and extracted minutiae points (**e**,**f**).

**Figure 7 sensors-22-08229-f007:**
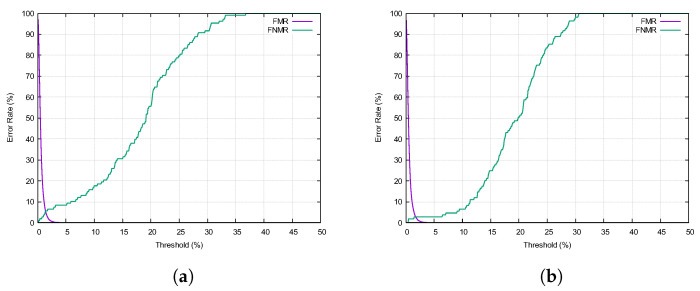
Comparison of OCT fingerprints and plain 2D fingerprints. (**a**) Outer fingerprint. (**b**) Inner fingerprint.

**Table 1 sensors-22-08229-t001:** Computation times on 3rd gen. AMD Ryzen and Nvidia RTX 2060.

Operation	Execution Time in ms
Copy volume from CPU to GPU memory	175
Volume flattening (GPU)	23
Outer fingerprint segmentation (GPU)	260
Inner fingerprint segmentation (GPU)	290
Surface masking (CPU)	20
2 × surface envelope (CPU)	50
2 × 3D-to-2D conversion (CPU)	4
Total time	822 ms

## Data Availability

In compliance with the General Data Protection Regulation, the data collected in this study are not publicly available.

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
