# Peer review of "Robust Identification and Segmentation of the Outer Skin Layers in Volumetric Fingerprint Data"

_sensors, 2022, doi:10.3390/s22218229_

Round 1

Reviewer 1 Report (New Reviewer)

General comments

1. Line 6. The OCT part of the abstract does not need to be introduced too much.

2. Line 12. The content of the abstract only focuses on the background and methods, and it is hoped that the test results, discussions, and conclusions can be added.

3. Line 21. In the introduction part, I hope to add more references to support the author's argument. It is unreasonable that the introduction of biological features in the first paragraph is not quoted at all

4. Line 29. The commercial fingerprint scanner is not easy to use. Please add a reference. If it is the author's statement, please explain the reason and provide verification

5. Line 32-35. Please add the reference to the description of the baby

6. Line 52. 2.1. OCT Basics, Please delete the introduction about OCT, or use a short sentence to introduce it, which is not very meaningful to the article. And I like to add references.

7. Line 106. 2.2. Anatomy of Fingertip Skin, this part of the literature is recommended to be incorporated into the first chapter, so there is no need for too much introduction

8. Line 131. 3. Related Work, This part of the literature is recommended to be incorporated into the first chapter, so there is no need for too much introduction

Specific comments

1 At present, fingerprint recognition is a stable technology, but the current counterfeiting is relatively expensive for OCT. The author seems not convincing enough to apply OCT to fingerprint recognition.

2 This article looks like the first three chapters of the dissertation. The paper's results, discussions, and conclusions have not been fully demonstrated. The author needs to complete the supplement and resubmit it.

Author Response

Dear reviewer,

thank you very much for taking the time to review our paper. I would like to address your comments:

General Comments

4. Using the commercial scanner is actually quite easy. For a good scan result, however, the conditions must be right (e.g. skin not too moist or dry), which is well known in the field. I will try to add sources that support the statement.

6/7/8. As you rightly suspect, the paper is part of a dissertation, which is cumulative, i.e. several papers are later put together fairly unchanged. In this paper, the first chapters therefore have to be lengthy. I can understand the criticism, but I see no way to compress all the information into the first chapter. The current chapter structure was chosen so that the reader is free to skip these sections and can still understand the core of the work.

All other points: No objections

Specific comments

The aim of this paper is to provide a method for segmenting backward compatible fingerprints from OCT data. The focus is therefore on the algorithms and how they were made sufficiently fast for practical use.

The comparison between OCT fingerprints and conventional 2D fingerprints has only been added to legitimize the output of the segmentation method. Of course, this is not a final treatment of the topic, but rather the first step.

You stated that "The author seems not convincing enough to apply OCT to fingerprint recognition." It remains to be seen whether OCT fingerprint recognition actually makes sense despite its costs. However, such proof would have gone beyond the scope of this paper.

As indicated in the summary, we are already advancing our research and development on this topic. Among other things, we are investigating the possibility of an intrinsically secure PAD. Depending on our results, we may be able to make a more compelling case for OCT for fingerprint recognition in a follow-up paper.

Next steps

The other 3 reviewers have already given their approval. Therefore, if possible, we would like to avoid extreme alterations that would need to be reconciled with the other reviewers. Would you agree with the following revisions?

  • Improved abstract
  • Addition of further sources where suggested
  • Improved result presentation (but it's unclear what exactly you expect, please elaborate)

Thanks again for your efforts. We look forward to your reply.

Reviewer 2 Report (New Reviewer)

Please the authors recheck the terminology (the Norwegian University of Science and Technology (NTNU) in this paper.

Author Response

Dear reviewer,

thank you very much for taking the time to review our paper.

As per your suggestion we have corrected the terminlology.

Best regards.

Reviewer 3 Report (New Reviewer)

In this paper, using Optical Coherence Tomography (OCT), a fast and robust digital signal processing circuit is proposed to segment two complementary fingerprints from the same OCT scan of the fingertip: One is taken as usual from the epidermal surface ("external fingerprint"), while the other is segmented from within the skin, right at the junction between the epidermis and the underlying dermis ("internal fingerprint").

The research was systematic and organized. Relevant study-related articles are linked, and the OCT Capture Device and feature set-up were described in detail.

In summary, the manuscript is well put together, and contains a lot of interesting new data, and therefore will be of interest to other researchers working in this field as well as readers of this journal.

Author Response

Dear reviewer,

thank you very much for taking the time to review our paper and your encouraging comments.

Best regards.

Reviewer 4 Report (New Reviewer)

The manuscript presents an Optical Coherence Tomography (OCT) technology-based fingerprint identification and segmentation algorithm. The manuscript is well organized. However, there is a point to be revised.

The algorithm works well in scenarios with consistent inner and outer fingerprint features. Can the system distinguish actual fingers and fingers with a thin film imitating a specific fingerprint? Some discussion about the potential of artificial fingerprint identification should be carried out.

Author Response

Dear reviewer,

thank you very much for taking the time to review our paper.

The topic you refer to is presentation attack detection (PAD), where OCT could offer a decisive advantage over conventional fingerprinting methods. We have deliberately not included this discussion, as it would have required the definition of attack scenarios, the creation of artefacts (i.e. fake fingers) and the development and testing of PAD algorithms, which would have been beyond the scope of this paper. However, as you rightly point out, the topic is important, which is why we plan to publish a follow-up paper that will address this question.

Again, thank you for your efforts.

Best regards

Round 2

Reviewer 1 Report (New Reviewer)

General comments

1. Line 6. The OCT part of the abstract does not need to be introduced too much.

2. Line 12. The content of the abstract only focuses on the background and methods, and it is hoped that the test results, discussions, and conclusions can be added.

3. Line 21. In the introduction part, I hope to add more references to support the author's argument. It is unreasonable that the introduction of biological features in the first paragraph is not quoted at all

4. Line 29. The commercial fingerprint scanner is not easy to use. Please add a reference. If it is the author's statement, please explain the reason and provide verification

5. Line 32-35. Please add the reference to the description of the baby

6. Line 52. 2.1. OCT Basics, Please delete the introduction about OCT, or use a short sentence to introduce it, which is not very meaningful to the article. And I like to add references.

7. Line 106. 2.2. Anatomy of Fingertip Skin, this part of the literature is recommended to be incorporated into the first chapter, so there is no need for too much introduction

8. Line 131. 3. Related Work, This part of the literature is recommended to be incorporated into the first chapter, so there is no need for too much introduction

Specific comments

1 At present, fingerprint recognition is a stable technology, but the current counterfeiting is relatively expensive for OCT. The author seems not convincing enough to apply OCT to fingerprint recognition.

2 This article looks like the first three chapters of the dissertation. The paper's results, discussions, and conclusions have not been fully demonstrated. The author needs to complete the supplement and resubmit it.

Author Response

Dear reviewer,

once again, thank you for reviewing our manuscript. Please find our reply below:

General comments

1. / 2. The abstract has been restructured

3. Relevant references added

4. Maybe I misunderstand the comment, but no statement was made about the commercial scanner being easy to use.

5. References added

6. Reference added

7. The information in this chapter is relevant to the concept of OCT fingerprinting.
If it were included in the introduction, the chapter would become too long or else information would have to be omitted.

8. As with point 7. the first chapter would then be overly long. Chapter 3 has no repetitive introduction, so there is no overhead.

Specific Comments

1. In this work, we focused on segmenting the required information from the OCT data and were able to demonstrate the validity of the results.
We will deal with counterfeiting in our future work, as this is a topic in its own right.

2. "This article looks like the first three chapters of the dissertation."
That is correct. The paper will be the first in a series that will be compiled into a dissertation.

"The paper's results, discussions, and conclusions have not been fully demonstrated."
This statement is unclear. As mentioned earlier, the focus of this work is on fingerprint segmentation. For this, we describe an algorithmic method and demonstrate its validity by cross-checking the results with plain 2D fingerprints.

Reviewer 3 Report (New Reviewer)

Accepted manuscript in present form

Author Response

Dear reviewer,

once again, thank you for your efforts.

Best regards.

This manuscript is a resubmission of an earlier submission. The following is a list of the peer review reports and author responses from that submission.

Round 1

Reviewer 1 Report

The paper approaches fingerprints biometrics using OCT. A digital processing is developed to use 3D images for the extraction of usable 2D images for fingerprints. The topic is of interest and the technical part of the study looks feasible, therefore the paper could be considered for publication in Sensors, with some necessary improvements, as suggested bellow.

1) The English must be corrected, grammar errors exist, words are missing here and there (Line 11: ‘of OCT’, not ‘OCT’, etc.), phrases are much too long (starting with the Abstract), style must be polished (e.g., do not use ‘publication’ but ‘work’ or ‘study’, etc.).

2) The Abstract must be better structured, with ‘aim of the work’ clearly stated, results more detailed, conclusions and relevance.

3) The Intro is much too brief in discussing the domain and the different alternative approaches. The proposed solution must be described in a convincing way starting with the Intro in comparison to other OCT-based solutions.

4) The refs are incomplete and not properly formatted, please correct. Also, there are too many conference papers, as well as groups of papers from the same authors. Should be improved.

5) OCT works should be completed, including with other reviews on OCT

Drexler, W., Fujimoto, J. G. State-of-the-art retinal optical coherence tomography. Progress in Retinal and Eye Research 2008;27(1);45–88.

and the state-of-the-art 2 microns axial and lateral resolution of Gabor-Domain Optical Coherence Microscopy

  1. Cogliati, C. Canavesi, A. Hayes, P. Tankam, V.-F. Duma, A. Santhanam, K. P. Thompson, and J. P. Rolland, MEMS-based handheld scanning probe with pre-shaped input signals for distortion-free images in Gabor-Domain Optical Coherence Microscopy, Optics Express 24(12), 13365-13374 (2016).

6) This reader does not see the use of presenting in Fig. 1 TD-OCT, this is history. LL or FF OCT might be interesting, though, although really useful is to focus on the system utilized in this work. Fig. 4, however, must be completed with a schematic of the setup (not only photos, they are not very useful for readers), even if it is a Thorlabs-based system.

7) Section 3 should be divided between Intro and 2.1, please complete OCT system description before going to skin. Please reorganize the manuscript in Intro, M&M, Results, Discussion and Conclusions, with appropriate subsections, according to the journal’s template and style.

8) The Discussion section must be added to the manuscript to demonstrate improvements of the proposed method with regard previous works, for example those of Sousedik, cited in relationship with the algorithms.

9) Refs must be given for all equations, or Appendix to deduce them.

In conclusion, the manuscript must be reorganized, completed and improved quite a lot to be considered for publication.

Author Response

The paper approaches fingerprints biometrics using OCT. A digital processing is developed to use 3D images for the extraction of usable 2D images for fingerprints. The topic is of interest and the technical part of the study looks feasible, therefore the paper could be considered for publication in Sensors, with some necessary improvements, as suggested bellow.

  • First of all, thank you for your effort in reviewing our paper. With this, we would like to respond to your feedback.

1) The English must be corrected, grammar errors exist, words are missing here and there (Line 11: ‘of OCT’, not ‘OCT’, etc.), phrases are much too long (starting with the Abstract), style must be polished (e.g., do not use ‘publication’ but ‘work’ or ‘study’, etc.).

  • Since we were not able to understand the criticism in all points, we have had the text proofread by a native speaker in the meantime. Apart from a few missing commas, the proofreader did not make any significant changes. However, we have tried to correct the mistakes you noted and have revised some of the wording.

2) The Abstract must be better structured, with ‘aim of the work’ clearly stated, results more detailed, conclusions and relevance.

  • We have rewritten the abstact, now stating more clearly the contents of the paper.

3) The Intro is much too brief in discussing the domain and the different alternative approaches. The proposed solution must be described in a convincing way starting with the Intro in comparison to other OCT-based solutions.

  • The intro is only intended to give the motivation for OCT, why the technology could be interesting for biometrics.
  • A detailed overview of other solutions can be found later under "Related Work", where we have listed the most important research groups in this field.

4) The refs are incomplete and not properly formatted, please correct. Also, there are too many conference papers, as well as groups of papers from the same authors. Should be improved.

  • Most of the related work has been done by a few major research groups, and the papers we reference are the starting point for our work.

5) OCT works should be completed, including with other reviews on OCT

Drexler, W., Fujimoto, J. G. State-of-the-art retinal optical coherence tomography. Progress in Retinal and Eye Research 2008;27(1);45–88.

and the state-of-the-art 2 microns axial and lateral resolution of Gabor-Domain Optical Coherence Microscopy

    Cogliati, C. Canavesi, A. Hayes, P. Tankam, V.-F. Duma, A. Santhanam, K. P. Thompson, and J. P. Rolland, MEMS-based handheld scanning probe with pre-shaped input signals for distortion-free images in Gabor-Domain Optical Coherence Microscopy, Optics Express 24(12), 13365-13374 (2016).

  • Thanks for the interesting sources, but we don't see any relevance to our work since we are using a commercial product and not developing it.

6) This reader does not see the use of presenting in Fig. 1 TD-OCT, this is history. LL or FF OCT might be interesting, though, although really useful is to focus on the system utilized in this work. Fig. 4, however, must be completed with a schematic of the setup (not only photos, they are not very useful for readers), even if it is a Thorlabs-based system.

  • For the described algorithms, it is actually almost irrelevant how the scan data is generated in the first place. Nevertheless, for the sake of completeness, we had decided give a brief explanation of OCT, which is most easily explained starting with TD.
  • Admittedly, Fig. 1 TD is not strictly needed. In our opinion, however, it helps an uninformed reader better understand how things work.
  • The OCT system used in this work is based on FD, which is described here. Furthermore, our scanner is basically a black box that we cannot reasonably discuss due to lack of information.
  • 4: We have replaced the photo of the measurement setup and added a screenshot ist CAD design where one can see how on the inside the scan head is mounted to the linear stage mentioned in the text.

7) Section 3 should be divided between Intro and 2.1, please complete OCT system description before going to skin. Please reorganize the manuscript in Intro, M&M, Results, Discussion and Conclusions, with appropriate subsections, according to the journal’s template and style.

  • From our understanding, the template structure is not mandatory and we would basically have to rewrite the entire paper to match it.

8) The Discussion section must be added to the manuscript to demonstrate improvements of the proposed method with regard previous works, for example those of Sousedik, cited in relationship with the algorithms.

  • Added some discussion to the later chapters.
  • A direct comparison of our work with that of Sousedik or others would be dubious, since we do not share a common database.
  • However, we were able to gain access to their scanner, and thus verify that our method also works with their data.
  • For now, this is the best we can do to legitimize our method (which the other authors have not done, probably for similar reasons)

9) Refs must be given for all equations, or Appendix to deduce them.

  • The equations are application-specific design decisions that are motivated in the text. Together with the algorithms, the method is described in such a way that it could be rebuild by the reader.

Reviewer 2 Report

In this paper, the authors proposed segmentation of the outer skin layers in volumetric fingerprint data and 2D finger image generation methods. Descriptions of devices and algorithms are well expressed and organized. However, it is considered that some verifications are lacking.

- The title of the paper contains the word "identification". However, verification of identification as a biometric method is not included.

- In the abstract, the problem of the existing method for presentation attacks limitation was mentioned. However, in this paper, it is not well expressed which part of the proposed method can overcome the limitation.

- Fig. 7 shows the result of converting to a 2D gray scale image and the detection result of the minutiae points. However, performance comparison with existing 2D fingerprint images is not included.

Author Response

In this paper, the authors proposed segmentation of the outer skin layers in volumetric fingerprint data and 2D finger image generation methods. Descriptions of devices and algorithms are well expressed and organized. However, it is considered that some verifications are lacking.

  • First of all, thank you for your effort in reviewing our paper. With this, we would like to respond to your feedback.

- The title of the paper contains the word "identification". However, verification of identification as a biometric method is not included.

  • The term identification refers to the fingerprint positions, not their subsequent use for identification purposes. Since this seems to be misleading, the term should perhaps be exchanged for "detection" or something along those lines.

- In the abstract, the problem of the existing method for presentation attacks limitation was mentioned. However, in this paper, it is not well expressed which part of the proposed method can overcome the limitation.

  • It was not our intention to discuss PAD in this paper. What was meant was that PAD could later be built on top our method. We have reworded the relevant passages to make this clearer.

- Fig. 7 shows the result of converting to a 2D gray scale image and the detection result of the minutiae points. However, performance comparison with existing 2D fingerprint images is not included.

  • Unfortunately, this was not reasonably possible due to a lack of data (covid related) and still unsolved issues (e.g. fingerprint unrolling) which are beyond the scope of this work.
  • We have added a description of our data base and tested our method with data from another scanner to proof its robustness.

Round 2

Reviewer 1 Report

The authors provided a revised version of their paper, but most of the comments (which were minimal) have not been actually properly considered. Rejecting improvement suggestions from reviewers is not a good response strategy. We suggest the authors to carefully consider all comments and queries made in the first review and to revise the manuscript accordingly. Until then, although the developed algorithms may be of interest, this reader cannot give but the same recommendation, of ‘Major revision’.

Reviewer 2 Report

The author presented a revised manuscript based on review comments. However, it is difficult to review the revised part only with the contents of the reply letter.
In addition, we did not respond to issues raised in relation to important experiments. Lack of data collection due to the Covid situation may be a reason, but it does not enhance the completeness of the paper.
Contributions can be recognized only when experiments to prove the utility of 3D fingerprint data obtained by the proposed method are supported.
Advantages in terms of utilization, such as better detection of minutiae points and robustness against spoofing attacks, should be verified.